# Nutritional Content of Popular Menu Items from Online Food Delivery Applications in Bangkok, Thailand: Are They Healthy?

**DOI:** 10.3390/ijerph20053992

**Published:** 2023-02-23

**Authors:** Nongnuch Jindarattanaporn, Inthira Suya, Lara Lorenzetti, Surasak Kantachuvesiri, Thaksaphon Thamarangsi

**Affiliations:** 1International Health Policy Program (IHPP), The Ministry of Public Health, Nonthaburi 11000, Thailand; 2Better Health Program (BHP) Thailand, Family Health International (FHI360), Bangkok 10330, Thailand; 3Global Health and Population Research, FHI 360, Durham, NC 27701, USA; 4Nephrology Division, Department of Medicine, Faculty of Medicine, Ramathibodi Hospital, Mahidol University, Bangkok 10400, Thailand; 5Thai Low Salt Network, Nephrology Society of Thailand, Bangkok 10310, Thailand

**Keywords:** nutritional content, online food delivery, popular menu items, Thailand

## Abstract

The rise in online food delivery (OFD) applications has increased access to a myriad of ready-to-eat options, which may lead to unhealthier food choices. Our objective was to assess the nutritional profile of popular menu items available through OFD applications in Bangkok, Thailand. We selected the top 40 popular menu items from three of the most commonly used OFD applications in 2021. Each menu item was collected from the top 15 restaurants in Bangkok for a total of 600 items. Nutritional contents were analysed by a professional food laboratory in Bangkok. Descriptive statistics were employed to describe the nutritional content of each menu item, including energy, fat, sodium, and sugar content. We also compared nutritional content to the World Health Organization’s recommended daily intake values. The majority of menu items were considered unhealthy, with 23 of the 25 ready-to-eat menu items containing more than the recommended sodium intake for adults. Eighty percent of all sweets contained approximately 1.5 times more sugar than the daily recommendation. Displaying nutrition facts in the OFD applications for menu items and providing consumers with filters for healthier options are required to reduce overconsumption and improve consumer food choice.

## 1. Introduction

Each year, noncommunicable diseases (NCDs) are responsible for 41 million deaths globally [1]. In Thailand, NCDs cause 75% of all deaths, with cardiovascular diseases (CVDs) accounting for the highest proportion [2]. Dietary factors, including increased intake of salt, fats, and sugars, are the biggest contributor to CVD risk [3]. Transnational food and beverage corporation practices have reshaped the dietary landscape through a combination of food availability, pricing, and social and cultural desirability [4], all of which have made unhealthy foods more readily accessible. In particular, food and beverage businesses have expanded their service channels to online food delivery (OFD) applications in order to provide convenience for consumers, a strategy which has also led to increased product sales [5]. The proliferation of OFD applications has provided a broader portion of the Thai population with direct access to non-traditional and ready-to-eat foods, which have the potential to disrupt good health and well-being [6].

OFD applications are currently considered a significant predictor of food choice [7] and eating [8] among the general population. The Thai food delivery market has grown rapidly, expanding from 61,000 million baht in 2019 to 68,000 million baht during the COVID-19 pandemic in 2020 [9], and further still to 105,000 million baht in 2021 [10]. The percentage of foods ordered from OFD applications increased from 3.9% to 10.7% between 2019 and 2020 [11]. In 2020, 85% of Thai people utilised OFD applications, with 61% of that group ordering fast food, such as fried chicken, burgers, and pizzas [12]. Use of OFD applications has grown more popular than restaurant dining or takeout among Thai people due to the convenience of searching for food items and finding new restaurants through these applications [13]. Food delivery trends in Thailand are similar to those found in other countries. Evidence from Australia, New Zealand, Canada, the United States, and the Netherlands have shown that most menu items, including the most popular items on OFD applications, were unhealthy [14,15,16,17] because of their high levels of salt, sugar, and/or saturated fats [14,15,16].

Raising public awareness about dietary guidelines and package labelling are some of the most common strategies utilised to educate the public about healthy diets [18]. Thailand established government policies to tackle unhealthy diets, specifically for packaged foods. Interventions such as the Guideline Daily Amounts (GDAs) label and “Healthier Choice” nutritional logos on selected packaged food products have been used to raise awareness about healthy eating [19,20]. However, these interventions only apply to packaged foods and do not encompass OFD applications. Although a previous study has assessed the nutrition information displayed on ready-to-eat packaged foods and the nutritional quality of those food products in Thailand [21], no data currently exists on the nutritional content of foods offered through OFD applications in Thailand. This study aims to address this gap by exploring the nutritional profile of popular menu items (food and non-alcoholic beverages) available through OFD applications. The goal is to raise public awareness about the nutritional content of foods delivered through these services and inform ongoing policy development and implementation for tackling unhealthy diets and NCDs in Thailand.

## 2. Materials and Methods

We conducted a cross-sectional, exploratory study to describe the nutritional contents of the most popular food and drink items available on OFD applications in Bangkok, Thailand. We summarised the nutritional contents from the 40 most popular menu items based on energy, total fat, sodium, and total sugar, and compared them against recommended daily intake values. This study received approval from FHI 360’s Office of International Research Ethics (report number 1892564-2).

### 2.1. Sample Selection

#### 2.1.1. Selection of Online Food Delivery Applications

Three OFD applications (Grab, Lineman, and Robinhood) were purposively selected due to their high cumulative utilisation rate among all OFD platforms; approximately 89% of Thai people used these applications when ordering their food and drinks through an OFD application [12]. Furthermore, they have consistently maintained their positions as leaders in Thailand’s OFD market [22]; Grab had the highest market share (50%), followed by Lineman (20%), and Robinhood (7%) [23].

#### 2.1.2. Identification of Popular Menu Items

Data on the most popular menu items from Grab, Lineman, and Robinhood were compiled and saved in Microsoft Excel between May and June 2021 [24,25]. Each application had its own list of most popular items, with each list differing slightly due to varying consumer preferences. All popular menu items across the three applications were selected for a total of 80 menu items. Next, 20 menu items were removed due to duplication. The remaining 60 menu items consisted of 20 items from Grab (33% of total menu items), 21 from Lineman (35%), and 19 from Robinhood (32%). However, given budget constraints for food nutrition analysis, the target population was reduced to 40 menu items. These items were selected based on their popularity ranking in each OFD application while maintaining the same proportion of items from the original sample size. Therefore, the target population comprised the top 13 items from Grab, the top 14 from Lineman, and the top 13 from Robinhood (Figure 1).

### 2.2. Data Collection

After identifying the most popular items across the three applications, Grab was ultimately used as the sole OFD application to order these items for data collection as it is the most popular OFD application in Bangkok [25,26,27,28]. The final 40 menu items were categorised into three different types: 25 ready-to-eat items, 5 sweets, and 10 non-alcoholic beverages. Research team ordered each menu item from each restaurant from the top 15 restaurants in Grab as nominated by consumers. For consistency, the order time was set between 8.00 a.m. and 12.00 p.m. during standard restaurant operating hours. The research team selected one standard portion of each menu as the default. Since restaurants have varying portion sizes, the research team recorded the weight of each sample to calculate a portion size average for each item and ensure more accurate results. All items were ordered within a one-month period (4 January to 1 February 2022). Delivery drivers for OFD applications delivered menu items to a laboratory, and each menu item was tested for nutritional content the day it was received (minimum of 500 g of sample needed).

### 2.3. Data Analysis

Each menu item’s nutritional contents were evaluated in terms of energy, total fat, sodium, and total sugar, as overconsumption of these nutritional contents is one of the risk factors associated with NCDs [29,30,31]. We opted to evaluate total fat instead of saturated fats due to budget and time constraints. Nutritional analysis for the items was conducted by Central Laboratory Co., Ltd., Bangkok, Thailand [32], with nutritional contents classified using chemical analysis [33]. The research team summarised the average, minimum, and maximum values of each item’s nutritional profile. SPSS version 26 was used for analysing the variation of nutritional content among the menu items.

The outcomes of this study were compared with national and international standards, as listed in Table 1. Since recommendations for total fat and total sugar intake are calculated on a daily basis, the research team calculated the recommended intake per portion for total fat and total sugar. This entailed dividing the daily recommended intake by three based on the assumption that one portion is equivalent to one meal and there are three meals in a day. Menu items with contents higher than the recommended criteria were categorised as “unhealthy menu items”.

## 3. Results

### 3.1. Nutritional Composition per 100 Grams

Overall, 40 menu items from 15 restaurants in Bangkok were classified into three food types: 25 ready-to-eat items, 5 sweets, and 10 non-alcoholic beverages.

Table 2 shows the nutritional content per 100 g for each of the 40 items. Overall, fried streaky pork and grilled pork neck were extremely high in energy and total fat per 100 g compared to other ready-to-eat items. Fried streaky pork had the highest energy and total fat (mean energy = 440.5 per 100 g; mean total fat = 36.3 per 100 g), followed by grilled pork neck (374.5 g and 30.8 g, respectively). In terms of sodium content, spicy papaya salad with northeastern style fermented crab and fish was especially high in sodium (1.6 g per 100 g)—nearly equivalent to the daily recommended maximum sodium threshold of 2 g. Grilled pork balls (0.8 g per 100 g) and grilled pork (0.8 g per 100 g) were also high in sodium. In terms of sugar, pandan and coconut chiffon cake ranked highest in total sugar (23 g per 100 g), followed by iced honey lemon tea (19.7 g per 100 g), and iced cocoa (16.2 g per 100 g).

### 3.2. Nutritional Content of Items by Portion and Comparisons with the WHO Daily Intake Standard

#### 3.2.1. Energy

Figure 2 illustrates the energy content of all 40 menu items. Fried streaky pork contained the highest average energy content per portion (814.9 kcal), followed by grilled pork (811.5 kcal), and rice with stir-fried minced pork, chili, and basil (734.4 kcal). Among sweets, pandan and coconut chiffon cake was the highest in average energy (1098.8 kcal), followed by egg tart (678.5 kcal). For non-alcoholic beverages, bubble milk tea was the highest in average energy (417.9 kcal), followed by iced green tea Frappuccino (382.2 kcal) and iced coffee (336.5 kcal).

The WHO’s recommended average daily energy requirement for adults is 2100 kcal per person per day, i.e., not more than 30% of recommended daily total energy [34]. Fried streaky pork contained an average of 814.9 kcal per portion (around 39% of total daily intake), and the pandan and coconut chiffon cake (307 g) provided an average of 1099 kcal per portion (around 50% of total daily intake). Bubble milk tea had an average 418 kcal per portion (around 20% of total daily intake).

Additionally, among all menu items, 7 were categorised as “unhealthy” in terms of energy content (five ready-to-eat foods and two sweets). The most “unhealthy” item was fried streaky pork followed: grilled pork; grilled pork neck; grilled pork balls; papaya salad, spicy, with dried shrimp and roasted peanuts; papaya salad, spicy, with fermented crab and fermented fish northeastern style; pandan and coconut chiffon cake; and egg tart. Notably, none of the non-alcoholic beverages fell into the “unhealthy” category.

#### 3.2.2. Total Fat

Six ready-to-eat items and three sweets contained higher fat than the WHO’s recommendation, but none of the non-alcoholic beverages were above the recommended threshold. The average total fat content per portion was highest for fried streaky pork (67.1 g), followed by grilled pork (55.6 g) and grilled pork neck (46.8 g). For sweets, the average total fat content per portion was greatest for pandan and coconut chiffon cake (65.1 g), followed by egg tart (45 g) and coconut milk ice cream (30.9 g) (Figure 3). Although these menu items consist of just one meal, they already contain nearly all of the WHO’s recommended total daily fat intake [34,35].

#### 3.2.3. Total Sodium

Overall, 8 out of 25 ready-to-eat items were very high in sodium (as measured by the daily sodium intake threshold of 2 g) and 23 of 25 ready-to-eat “unhealthy” menu items contained more than the recommended sodium intake for adults of 0.6 g per meal (Figure 4). For reference, the WHO suggests that a person should consume less than 5 g of salt (approximately 2 g sodium) per day [36].

Mean sodium levels were much higher when reported per portion rather than per 100 g. The average total sodium content per portion was greatest for spicy papaya salad with fermented northeastern style crab and fish, Chinese pork bun, and iced coffee. One portion of spicy papaya salad with fermented northeastern style crab and fish (313 g) contained 5 g of sodium, and the average portion for Chinese pork bun contained 0.8 g of sodium. High sodium was not only found in ready-to-eat items but also in non-alcoholic beverages. Iced coffee was found to have the highest amount of sodium per portion among non-alcoholic beverages at 0.3 g.

#### 3.2.4. Total Sugar

Eight non-alcoholic beverages were considered unhealthy (more than 25 g of sugar per portion) (Figure 5). The WHO recommends that adults and children reduce their daily intake of free sugars from less than 10% of their total energy intake to less than 5%, or roughly 25 g (6 teaspoons) per day [35,37,38,40]. All non-alcoholic beverages, except for soy milk and iced Americano, contained an average of 33.9 g of sugar per portion, and all sweets except for egg tart and deep-fried Chinese dough contained an average of 31.5 g of sugar per portion; this is almost 1.5 times higher than the daily recommendation.

Notably, the average sugar content per menu item may not be indicative of whether a certain item is “healthy” or “unhealthy” in the Thai context when compared to the WHO’s recommendation. Although the average sugar content of an item may show that it is “healthy”, this is also based on the average portion size and the standardisation of ingredients. Thus, if a certain item’s portion size happens to be much larger than the average, or if a certain restaurant’s recipe uses more sugar than normal, it is possible that the item may be categorised as “unhealthy”. For example, the average sugar content of rice with salmon was 7.7 g, which is considered “healthy”. However, the sugar content range for this menu item was 0–21.6 g, with the maximum value close to the WHO recommended daily sugar intake (25 g).

## 4. Discussion

Most of the menu items were considered unhealthy, with higher levels of energy, total fat, sodium, and total sugar compared to the recommended daily intake. The findings of this study correspond to similar studies in China and Canada where the nutritional quality of OFD foods was generally low [41] and did not meet healthy eating recommendations [16]. The nutritional information generated from analysing the 40 menu items can serve as a launching point for both practical actions in the form of regulating information provided through OFD applications and raising consumer awareness about nutritional contents. The large variations in total fat, sodium, and sugar content observed when comparing menu items per portion and per 100 g indicate that opportunities exist for improvement. This can be achieved by standardizing portion size or showing nutritional facts for menu items through the OFD applications, particularly for sodium, sugar, and fat. These approaches may reduce the overconsumption of unfavourable nutrients, and are strategies advocated for addressing NCDs [42].

### 4.1. Energy Content

When analysing these menu items, a portion of fried streaky pork delivered via OFD applications contained 39% of the WHO’s recommended daily energy intake for adults, and 37% and 46% of the recommended daily energy intake for Thai men and women aged 19 to 50 years, respectively, based on the Department of Health (DOH), Ministry of Public Health (MoPH). This does not account for any additional accompaniments, such as rice (one ladle) that can add approximately 80 extra calories [38]. Furthermore, the DOH recommends aiming for approximately 400–600 calories for a main meal [38]. Many menu items, including drinks, contain nutrients that are higher than the DOH recommendations for daily caloric intake. Restaurants should consider improving the overall nutritional profile of these items by reformulating the recipe or cooking method, or by reducing portion size.

### 4.2. Total Fat Content

Six ready-to-eat items and three sweets had higher fat content than the WHO’s and DOH’s recommended total fat intake, which is 20–35% (44–78 g) of total energy intake for Thai adults [35]. This is particularly problematic since desserts are likely to be consumed alongside a main meal, meaning consumers are consuming more fat than recommended in a single meal. Restaurants should consider substituting ingredients with lower fat alternatives. For example, since one of the main ingredients in pandan and coconut chiffon cake is high fat oil, bakery shops should consider substituting this with reduced fat oils.

### 4.3. Sodium Content

Sodium content was particularly high among the menu items assessed, which did not include any condiments that are often added to meals. WHO evidence revealed that Thais consume an average of 10.8 g of salt per day or 4.2 g of sodium in their current lifestyle, which was more than double the recommended daily amount of salt in 2015 [43]. A cross-sectional population-based survey conducted in Thailand in 2021 revealed that average sodium consumption among Thai adults was 3.6 g per day [44]. Our study supports this finding since many popular menu items in our analysis were also found to be high in sodium, and recipes with alternatives to sodium, such as low sodium condiments, were not popular due to higher prices. Thailand has set an ambitious goal of reducing the population intake of salt/sodium by 30% [45]; this is in line with the WHO’s global voluntary targets for a 30% relative reduction in mean population intake of salt/sodium by 2025 (relative to 2010 levels) [46]. Based on the WHO’s and DOH’s recommendations for daily and per meal sodium intake, restaurants should reduce sodium content by reformulating their recipes and providing nutritional information through OFD applications to enhance consumer awareness and transparency.

### 4.4. Sugar Content

All non-alcoholic beverages (except for soy milk and iced Americano) and all sweets (except for egg tart and deep-fried Chinese dough) contained average sugar content higher than the daily recommendation. A new WHO guideline recommends that ‘free’ sugars make up no more than 10% of daily kilojoule intake [37]. Notably, total sugar refers to the total amount of sugar from all sources (free sugars plus those from milk and those present in the structure of foods such as fruit and vegetables). Our nutritional analysis does not distinguish between naturally occurring sugars and free sugars. However, it is likely that the sugar content of the various papaya salads, pandan and coconut chiffon cakes, and iced honey lemon teas exceeded the WHO’s and DOH’s daily sugar recommendation for adults [35,37,38,40].

### 4.5. Policy Implications

Revising the Thai national policy could be another method for tackling sugar consumption. An updated excise tax has been applied to sugary drinks since 16 September 2017 [47]. The levy on sugary drinks is capped at 20%, with beverages containing more sugar carrying a larger tax burden than less sweet beverages [48]. However, this policy focuses on sweetened beverages in the form of packaged foods sold at retailers or supermarkets. The results of our study found that almost all sweets and non-alcoholic beverages are not categorised as packaged food since foods sold at restaurants are not required to be labelled. Despite their lack of inclusion in the policy, there is scope for restaurants to revise their recipes to reduce sugar content while concurrently displaying nutritional facts on OFD applications to help consumers make informed food choices that contribute to a healthy diet.

In addition to the policy strategies suggested above to reduce consumer intake of foods high in fats, sodium, and sugars, relevant public entities could collaborate or partner with OFD application developers to provide healthier food options. This can be accomplished in several ways. First, a voluntary upper limit could be set for sugar, fat, and sodium in the OFD applications or in restaurant menu details to indicate that the item is a “healthier” option. If a menu item is under the threshold, it can be indicated as “healthier”. Second, OFD application developers could design settings to allow consumers to filter options when they order. For example, they can choose to filter foods or restaurants by “less salt”, “less sweet”, and “less fat”. Third, public entities and developers could work together with restaurants to set and implement standardised portions for menu items available through the application. Finally, a logo can be designed for use on OFD applications to inform consumers that the food is healthy.

Restaurants should know whether the foods they are selling are unhealthy or not. The Bureau of Nutrition, MoPH has produced the Thai Nutri Survey Program (TNS) and the relevant manuals to address this issue. Therefore, social marketing should be used to promote this program among restaurants or public to raise awareness and provide the tools to analyse and monitor the nutritional content of their menu items. Consequently, restaurants will know how healthy their menu items are. This nutrition content should also be shown on the application to provide information for consumers. This will enable them to make informed food choices when ordering.

### 4.6. Strength and Limitations

To the best of our knowledge, this study is the first to investigate the nutritional content of popular menu items from OFD applications in Thailand. It analysed nutritional content with assistance from a professional laboratory, thus providing objective data results and helping to reduce the knowledge gap related to nutritional information for some of Thailand’s most popular foods and drinks. However, the study also has several limitations. First, this study only considered 15 restaurants in Bangkok that were ranked based on popularity by Grab and did not consider popular menu items from other applications or locations. In addition, popular menu items obtained through the OFD applications are only valid within the study period and may only be relevant to Bangkok. Therefore, these findings may not be relevant to popular menu items outside the study duration if recipes are changed, or to other parts of the country. However, because this study had wide-ranging results, adding more restaurants is unlikely to significantly affect the results. Second, budget constraints prohibited the addition of condiments into the analysis. Future studies should include condiments to provide more accurate results that represent a complete meal and improve understanding of typical consumption patterns. In addition, no data exists that compares home-cooked foods with OFD foods; it would be helpful to investigate whether the same menu items made at home are healthier.

Finally, the WHO’s most recent guidelines for daily energy, fat, salt, and sugar intake were used to evaluate salt and sugar levels in popular menu items. Although these guidelines are based on scientific evidence [46], limitations exist. The guidelines do not classify gender and age so average values may not be fully applicable in the Thai context due to differences in physiology between Thais and people of other races/ethnicities. Moreover, most menu items in this study did not meet international standards for energy and fat per meal. Further exploration is required to obtain a more accurate standard to assess the healthiness of foods.

## 5. Conclusions

OFD platforms are becoming popular, with an increasing number of orders for ready-to-eat foods, sweets, and non-alcoholic beverages. However, we found that most single items purchased through OFD applications in Bangkok contained levels of energy, total fat, sodium, and total sugars that were close to or exceeded recommended daily intakes. This creates additional challenges for public health nutrition policymakers, though OFD platforms may also provide an opportunity to improve public health nutrition and diet-related health outcomes using certain policy levers. It will be important for relevant entities under the MoPH—NCD Division and DOH—to collaborate with OFD application developers to use their influence and promote healthy food consumption. Such a public-private partnership may help increase the availability of healthy choices while also nudging consumers towards these options. Going forward, the nutritional contents of popular menu items should be randomly assessed. Condiments and other menu items from OFD applications not included in this assessment, as well as items from restaurants in other Thai provinces, should be included in future studies to increase the comprehensiveness of nutritional content measurement and analyses in Thailand.

## Figures and Tables

**Figure 1 ijerph-20-03992-f001:**
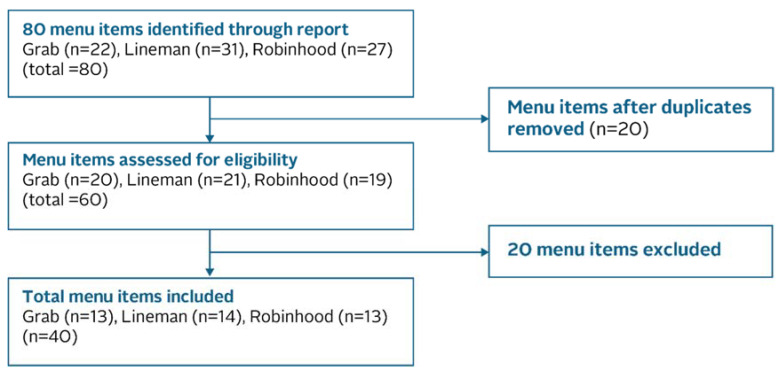
Sampling frame.

**Figure 2 ijerph-20-03992-f002:**
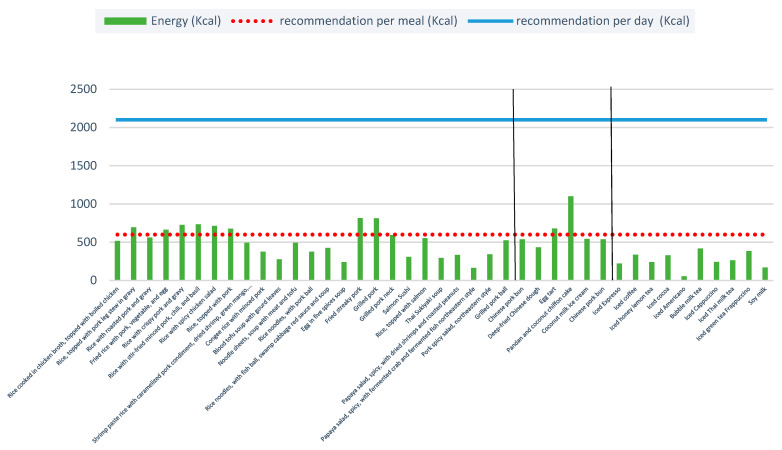
Total energy content per portion of the 40 menu items. Note: Two vertical blacks line separate the three food types: ready-to-eat items, sweets, and non-alcoholic beverages.

**Figure 3 ijerph-20-03992-f003:**
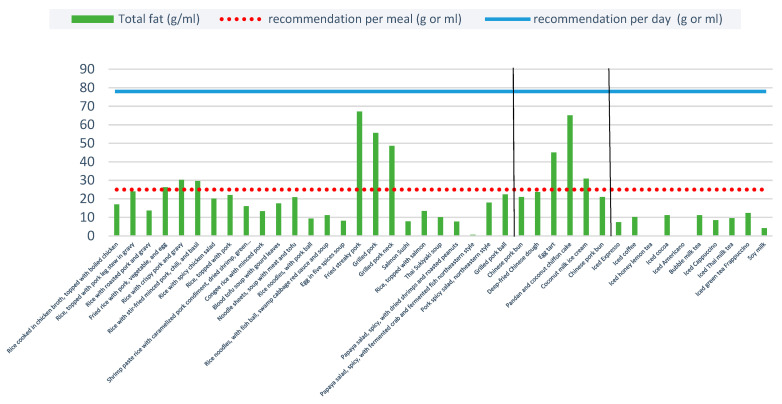
Total fat content per portion of the 40 menu items. Note: Two vertical blacks line separate the three food types: ready-to-eat items, sweets, and non-alcoholic beverages.

**Figure 4 ijerph-20-03992-f004:**
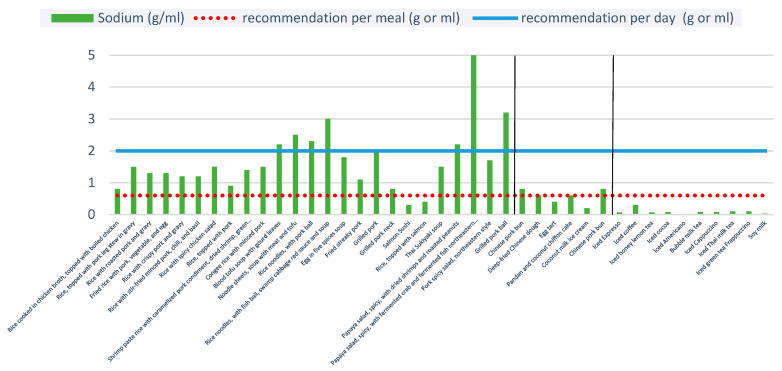
Total sodium content per portion of 40 menu items. Note: Two vertical blacks line separate the three food types: ready-to-eat items, sweets, and non-alcoholic beverages.

**Figure 5 ijerph-20-03992-f005:**
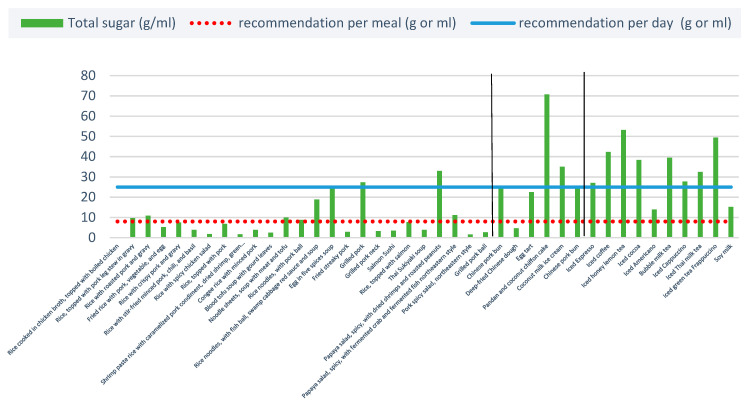
Total sugar content per portion of 40 menu items. Note: Two vertical blacks line separate the three food types: ready-to-eat items, sweets, and non-alcoholic beverages.

**Table 1 ijerph-20-03992-t001:** Daily nutritional intake recommendations according to the World Health Organization and Department of Health (DOH), Ministry of Public Health, Thailand, and adjusted intake recommendations per meal.

Threshold/ Source	Population	Energy	Total Fat	Sodium	Total Sugar
Daily/WHO and DOH	Adult	2100 kcal/person/day [34]	Not exceeding 30% of total energy intake fat intake (approximately 78 g) [35]	2 g sodium/person/day [36]	6 teaspoons/person/day (approximately 25 g) [37]
Per meal/Various sources		600 kcal [38]	600 mg [39]	8 g ^1^	25 g ^2^ [35,37,38,40]

^1^ Estimated by research team as a proportion of WHO standard of daily intake. This entailed dividing the daily recommended intake by three (based on the assumption that one portion is equivalent to one meal and there are three meals in a day). ^2^ Estimated by research team as a proportion of WHO standard of daily intake. This entailed dividing the daily recommended intake by three (based on the assumption that one portion is equivalent to one meal and there are three meals in a day).

**Table 2 ijerph-20-03992-t002:** Nutritional content of menu items.

Menu Items	Portion Weight g, mL (Min–Max)	Energy kJ/100 g (Min–Max)	Total Fat g/100 g (Min–Max)	Total Sodium g/100 g (Min–Max)	Total Sugar g/100 g (Min–Max)
Ready-to-eat items					
1. Rice cooked in chicken broth, topped with boiled chicken	291 (230–370)	177.2 (135.9–206.2)	5.8 (3.2–8.5)	0.3 (0.2–0.5)	0.0 (0.0–0.0)
2. Rice topped with pork leg stew in gravy	453 (350–600)	153.3 (133.8–218.7)	5.3 (3.1–10.9)	0.3 (0.2–0.5)	2.2 (1.6–3.1)
3. Rice with roasted pork and gravy	355 (150–475)	158.2 (133.9–200.9)	3.8 (4.2–22.8)	0.4 (0.2–0.7)	3.1 (1.6–6.6)
4. Fried rice with pork, vegetable, and egg	344 (260–425)	192.7 (156.3–235.9)	7.6 (4.2–11.7)	0.4 (0.2–0.7)	1.5 (0.0–2.7)
5. Rice with crispy pork and gravy	384 (300–540)	189.1 (152.2–231.6)	7.9 (4.1–14.0)	0.3 (0.2–0.4)	1.9 (0.0–6.6)
6. Rice with stir-fried minced pork, chili, and basil	397 (280–512)	185.0 (139.6–221.0)	7.4 (3.2–11.4)	0.3 (0.1–0.8)	0.9 (0.0–2.3)
7. Rice with spicy chicken salad	383 (290–475)	185.8 (151.2–237.4)	5.3 (3.9–9.0)	0.4 (0.3–0.6)	0.5 (0.0–2.7)
8. Rice topped with pork	377 (270–500)	179.1 (116.5–241.9)	5.9 (1.0–12.5)	0.2 (0.09–0.4)	1.8 (0.0–3.1)
9. Shrimp paste rice with caramelised pork condiment, dried shrimp, green mango, chilies, lime, shallot, cucumber, and long beans	248 (180–355)	198.7 (162.2–240.0)	6.4 (1.9–8.9)	0.6 (0.3–0.9)	0.7 (0.0–3.8)
10. Congee rice with minced pork	578 (490–760)	65.1 (52.5–88.0)	2.3 (1.2–5.6)	0.3 (0.2–0.4)	0.7 (0.0–1.7)
11. Blood tofu soup with gourd leaves	615 (502–730)	44.7 (23.0–104.4)	2.8 (1.0–9.8)	0.4 (0.2–0.5)	0.4 (0.0–2.2)
12. Noodle sheets, soup with meat and tofu	635 (480–800)	77.8 54.3–114.0	3.3 1.7–6.5	0.4 0.2–0.7	1.6 0.0–3.5
13. Rice noodles with pork ball	615 (508–842)	60.9 (36.3–80.4)	1.5 (0.6–2.3)	0.4 (0.3–0.5)	1.4 (0.0–3.3)
14. Rice noodles with fish ball, red sauce, and soup	628 (427–857)	67.5 (46.6–80.4)	1.8 (0.6–3.2)	0.5 (0.3–0.6)	3.0 (1.1–6.0)
15. Egg in five spices soup	281 (190–360)	85.3 (56.9–129.3)	2.9 (2.0–4.0)	0.6 (0.3–0.9)	8.9 (1.7–17.1)
16. Fried streaky pork	185 ^1^ (90–280)	440.5 (361.2–531.0)	36.3 (23.3–49.9)	0.6 (0.4–0.9)	1.6 (0.0–5.0)
17. Grilled pork	264 (100–420)	307.4 (255.1–347.5)	21.1 (13.5–27.0)	0.8 (0.5–1.3)	10.3 (3.9–14.7)
18. Grilled pork neck	158 (100–260)	374.5 (254.2–461.7)	30.8 (16.0–43.9)	0.5 (0.2–0.8)	2.1 (0.0–7.3)
19. Salmon sushi	177 (100–260)	173.5 (122.7–206.8)	4.4 (0.4–8.4)	0.2 (0.03–0.4)	1.9 (0.0–5.6)
20. Rice topped with salmon	311 (232–379)	177.6 (149.4–203.0)	4.3 (1.5–9.0)	0.1 (0.01–0.5)	2.5 (0.6–0.9)
21. Thai Sukiyaki soup	657 (515–820)	45.0 (30.6–63.5)	1.5 (1.0–2.2)	0.2 0.1–3.5	0.6 (0.0–1.7)
22. Papaya salad, spicy, with dried shrimp and roasted peanuts	302 (230–400)	110.4 (52.6–145.8)	2.6 (1.1–4.9)	0.7 (0.4–1.7)	10.9 (4.3–19.2)
23. Papaya salad, spicy, with fermented crab and fermented fish northeastern style	313 (240–140)	52.4 (38.9–72.9)	0.2 (0.1–0.5)	1.6 (1.3–2.1)	3.6 (0.0–8.5)
24. Pork spicy salad, northeastern style	241 (180–330)	142.2 (91.3–205.9)	7.5 (3.1–13.4)	0.7 (0.4–1.0)	0.7 (0.0–2.3)
25. Grilled pork balls	392 (268–548)	133.9 (86.8–190.1)	5.7 (1.7–11.6)	0.8 (0.4–1.1)	0.7 (0.0–2.6)
Sweets					
1. Chinese pork bun	213 (110–350)	252.6 (213.7–299.8)	9.9 (3.8–18.0)	0.4 (0.2–0.6)	11.5 (7.2–22.8)
2. Deep-fried Chinese dough	105 (40–200)	412.6 (343.8–492.8)	22.6 (12.3–33.3)	0.5 (0.3–0.9)	4.4 (1.1–9.0)
3. Egg tart	187 (90–270)	362.9 (287.5–475.2)	24.0 (16.9–32.3)	0.2 (0.1–0.3)	12.0 (7.1–15.4)
4. Pandan and coconut chiffon cake	307 (70–500)	357.9 (249.9–413.1)	21.2 (15.1–25.0)	0.2 (0.1–0.4)	23.0 (7.1–32.1)
5. Coconut milk ice cream	309 (140–660)	175.4 (120.2–213.5)	10.0 (3.6–14.5)	0.05 (0.09–0.1)	11.3 (6.1–16.0)
Non-alcoholic beverages					
1. Iced espresso	207 (160–260)	106.9 (82.7–128.9)	3.6 (2.3–4.7)	0.03 (0.02–0.04)	13.1 (9.7–16.3)
2. Iced coffee	250 (180–530)	134.6 (89.7–185.1)	4.1 (1.4–6.6)	0.3 (0.1–0.6)	16.9 (8.2–24.3)
3. Iced honey lemon tea	270 (185–360)	88.5 (53.7–113.1)	0.0 (0.0–0.1)	0.03 (0.002–0.02)	19.7 (13.3–24.3)
4. Iced cocoa	236 (200–320)	138.9 (97.9–206.7)	4.7 (3.2–6.6)	0.03 (0.03–0.04)	16.2 (10.3–28.4)
5. Iced Americano	214 (180–269)	24.9 (5.7–47.8)	0.1 (0.0–0.2)	0.004 (0.003–0.01)	6.5 (3.2–10.2)
6. Bubble milk tea	326 (250–460)	128.2 (86.1–186.8)	3.4 (1.6–5.3)	0.03 (0.01–0.04)	12.1 (7.1–18.3)
7. Iced cappuccino	220 (170–270)	110.6 (84.8–138.9)	3.9 (1.5–5.4)	0.04 (0.02–0.05)	12.6 (6.9–15.7)
8. Iced Thai milk tea	231 (174–294)	113.9 (79.3–157.6)	4.1 (2.7–7.3)	0.04 (0.03–0.06)	14.0 (7.2–26.5)
9. Iced green tea Frappuccino	563 (470–680)	67.9 (51.2–96.6)	2.2 (1.4–4.2)	0.02 (0.01–0.05)	8.8 (6.4–12.0)
10. Soy milk	355 (280–460)	47.6 (20.6–71.4)	1.2 (0.1–2.6)	0.009 (0.002–0.03)	4.3 (0.0–7.7)

Note: n = 15 for each menu item. Each menu item was ordered and measured nutritional contents 15 times. ^1^ Cells highlighted in pink in Table 2 indicate menu items that were extremely high in energy, total fat, total sodium, and total sugar per 100 g.

## Data Availability

This study analyzes quantitative data. The quantitative analysis table are available from the corresponding author on reasonable request.

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
