# Peer review of "Nutritional Content of Popular Menu Items from Online Food Delivery Applications in Bangkok, Thailand: Are They Healthy?"

_ijerph, 2023, doi:10.3390/ijerph20053992_

Round 1

Reviewer 1 Report

Introduction: 

-Paragraph 2: You argue the increase in OFP utilization and recognize COVID as a factor during this time period.  Do you have recent data on current rates (in 2022)? Are they still the same during peak COVID time? Have they gone down?

-How is OFD different than traditional sit-down or takeout options? We know that eating out is often less healthy than eating in. Is OFD being used more than eating out or getting takeout?

Methods:

-Why did you only focus on 2 months - May and June? Certainly food purchasing habits are different during 2 months of a similar season. 

- 8 AM to 12:00 PM? This is only for hours. This does not seem inclusive enough.  

- Analysis overall seems very limited in scope - only looking at energy, fat, sodium, and sugar. Additionally, only looking at total fat. 

Results:

-Overall, the results presented very clearly

-The pink higlighting in the table is not clear. Add a footnote to the tables.

Discussion:

-Again, I am asking myself how OFD is different than restaurant dining or takeout.  From a next steps perspective, how is this any different? What efforts are currently taking place for in restaurant dining or take out dining? This needs to be differentiated more to really understand the value of the research and how it differs from other approaches.  

Author Response

Please kindly see the attachedment.

Reviewer 2 Report

This study is really very useful and timely content.  I enjoyed reading it, and it is useful for policy advocacy as well. Saying this, my comments below:

Line 33-35

“Transnational 33 food and beverage corporation practices have reshaped the dietary landscape through a 34 combination of food availability, pricing, and social and cultural desirability (4), all of 35 which have made unhealthy foods more readily accessible.”

Not clear how it affected – more information please? And how it related to OFD?

line 69-70

This study received approval from FHI 360’s Office of Inter- 69 national Research Ethics.

Would you provide ethics number for the full information

Line 95

“Grab was ultimately used as the sole OFD application to order these items for data collection as it is 96 the most popular OFD application in Bangkok (26-29).”

It is not clear why Grab has been used as a sole OFD application for all the 40 items- why not each item ordered from the OFD platforms they originally selected from?  The rationale is not clearly justified to judge the content of nutrients of 40 meals just based on one application.

Line 99

“…the top 15 restaurants…”

Is this mean one or two item per restaurant? can you please clear on this?

Table 1

8 g1 and 25g2

Superscript 1 and 2 missed description as footnote?

Table 2

The footnote: “Note: n=15 for each menu item.”

-          Is this means you have ordered/measure contents 15 times for each food type? Please be clear on this.

-          Please clearly specify what the pink highlighted boxes indicate?  the highest nutrients content. Is this means the other items are as per national and international recommended range? - please specify as a foot note or more information in the text rather than just saying the y are the highest in each of the specified nutrients

Figures

You have presented all 40 items

-          Recommend an indicative line separating the three menu items (ready-to-eat, sweets, non-alcoholic bev) so that reader can easily understand which od the three categories are high in each nutrients

Line 258-261

The researchers recommended restaurants’ responsibility

but what about national policy recommendation (except from tackling sugar content)

Line 217

“To the best of our knowledge, this study is the first to investigate the nutritional con- 217 tent of popular menu items from OFD applications in Thailand.”

Suggest moving this sentence to the “strength and limitations of the study section- please refer my comment below as well.  

Line 304

Limitations

You have started with the strength of your study, so I recommend changing the title as “Strength and limitations?”

Line 305

“This study contains several strengths.”

However, I think only one strength is specified- please check. also see my comment above about moving the first sentence of discussion here.

Author Response

Please kindly see the attachement.

Reviewer 4 Report

The manuscript is well-designed and explained. However, it is suggested to make some minor revisions as follows:

Introduction:

A detailed literature review section should be added to support the research gap and novelty of this study.

Materials and Methods: 

A figure outlining the research framework would be helpful to understand the research design.

Results:

The meaning of color differences in Table 2 should be clearly explained. In general, each table and figure should be self-explanatory.

Discussion:

This section can be organized into several sub-sections with headings.

Conclusion:

The future direction of this study should be explained in this section.

Author Response

Please kindly see the attachement.
